# Health Promotion through Movement Behaviors and Its Relationship with Quality of Life in Spanish High School Adolescents: A Predictive Study

**DOI:** 10.3390/ijerph18147550

**Published:** 2021-07-15

**Authors:** Mikel Vaquero-Solís, Miguel Angel Tapia-Serrano, David Hortigüela-Alcalá, Manuel Jacob-Sierra, Pedro Antonio Sánchez-Miguel

**Affiliations:** 1Department of Didactics of Musical, Plastic and Body Expression, Faculty of Teaching Training, University of Extremadura, Avenida Universidad, S/N, 10071 Cáceres, Spain; mivaquero89@gmail.com; 2Department of Specific Didactics, Faculty of Education, University of Burgos, CalleVilladiego, 1, 09001 Burgos, Spain; dhortiguela@ubu.es; 3Physical Education Department, Faculty of Education, University of Castilla-La Mancha, Campus Universitario, S/N, 16071 Cuenca, Spain; jacobsierradiaz@hotmail.com

**Keywords:** high school, teachers, physical activity, health

## Abstract

A growing number of studies have highlighted the health benefits of high physical activity, low screen time, and optimal sleep duration among school-age children and adolescents. *Objective:* The present study proposes to examine the individual and combined association between movement behaviors (physical activity, screen time, and sleep time) and quality of life in boys and girls. *Method*: A total of 319 Spanish primary and secondary school students participated in the study. Physical activity, screen time, sleep duration, and quality of life were evaluated. *Results:* Descriptive, correlation, and regression analyses were carried out in order to improve knowledge about health-related behaviors for all participants. The results found significant positive associations between physical activities and sleep time with quality of life. Finally, the regression models showed that physical activity scores predict quality of life, especially in children. It is concluded that movement behaviors are important in association with quality of life. Likewise, the impact of physical activity on the quality of life is highlighted as the main behavior in the prediction of the quality of life for a population of school adolescents.

## 1. Introduction

There is an increasing number of studies that have highlighted the health benefits of high physical activity, low screen time, and optimal sleep duration among school-aged children [1,2,3]. Regarding these findings, some systematic reviews have indicated that children and adolescents with a healthy lifestyle had better overall health [4,5]. Specifically, they have revealed that school-aged children and youth with high physical activity [6], low screen time [7], and greater sleep duration [8] had better health indicators (e.g., physical and mental health and psychosocial well-being) than those in an unhealthy lifestyle.

Health-related quality-of-life is an important and multi-dimensional indicator for youth physical, mental, emotional, and social functioning [9,10]. Since health-related quality of life is an indicator of health, previous studies have examined their independent relation to movement behaviors (i.e., physical activity, screen time, and sleep duration) [11,12,13,14]. In this regard, it is important to evaluate movement behaviors from a holistic perspective in the 24 h period, since these have an important impact on health indicators [15]. Previous studies have linked quality of life with a higher level of physical activity [16], less screen time [16,17], and optimal sleep duration [12]. However, research has shown that these three behaviors are codependent and should be studied simultaneously [6,7,13,18,19].

The systematic review conducted by Saunders et al. [13] reported that the relationship between combinations of movement behaviors and health indicators has been little studied. In this regard, few studies have assessed the relationship between movement behaviors and health-related quality of life [20]. This study found a positive relationship between movement behaviors and health-related quality of life in adolescents. Despite these findings, mixed results have been reported between boys and girls in this relationship [20]. Specifically, the relationship between all three movement behaviors and health-related quality of life was significant in terms of the interaction of the three behaviors in girls, whereas the interaction of the three behaviors in boys points in the same direction, but was not significant compared to what was found in girls. In addition, greater importance was observed in relation to the quality of life for those participants who especially complied with the recommendations for sleep and sedentary time [20]. Given that there is no convincing evidence, more studies that examine the relationship combinations of movement behaviors with health-related quality of life are needed [13,20]. In this sense, 24-h movement behaviors represent a new paradigm under the idea that ‘all day matters’, according to Tremblay et al. [15]. This new paradigm, which is supported by the scientific literature, alludes to the importance of assessing movement behaviors 24 h a day compared to the paradigm that uniquely valued movement behavior related to health indicators (e.g., physical activity) [21]. Thus, the Canadian Guide’s line for movement behavior per 24 h establishes that children between 5 and 13 years old must comply with the recommendations of 9 to 11 h of sleep a day, accumulate 60 min of moderate to vigorous activity a day, several hours of a variety of structured and unstructured light physical activities, no more than 2 h of screen time, and no sitting for long periods of time [15].

Therefore, the school domain is a great context to promote those behaviors related to health, and teachers play an important role in promoting those healthy behaviors through active methodologies (i.e., Flipped Classroom, Gamification, etc.) [22]. For all the above, the present article attempts to answer the following research question: “Is there a significant interdependent relationship between the three movement behaviors and healthy lifestyle among boys and girls?” Thus, the purpose of this study was (i) to analyze the association between health-related quality of life with individuals and combinations of movement behaviors and (ii) to identify gender differences in this association. Although a possible relationship between the combinations of movement behaviors and health-related quality of life is expected, especially in girls, no hypotheses were formulated about the association of movement behaviors and quality of life related to health, or the possible differences in both boys and girls.

## 2. Materials and Methods

### 2.1. Participants

The present cross-sectional study was developed in Extremadura (Spain). The baseline data were collected before the COVID-19 pandemic. A total of 319 Spanish students from primary and high schools, including children (*n* = 150) and adolescents (*n* = 169), participated in the study. Individuals ranged in age from 10 to 14 years old (12.14 ± 1.23 years), including 163 boys (12.14 ± 1.24 years) and 156 girls (12.15 ± 1.23 years). The selection of the sample was carried out through an intentional sampling for convenience according to the distance of the schools to the research staff in charge of data collection, the willingness to collaborate on the part of the teaching staff, and the time required for the researchers to travel towards collaborating centers. Likewise, all subjects consented to their participation in the study.

### 2.2. Measures

Physical activity and screen time. Physical activity and screen time was measured using the Youth Activity Profile Questionnaire—Spain (YAP-S) [23]. This self-report instrument, designed to measure physical activity and screen time in youths (Saint-Maurice and Welk [24]) was validated in Spanish children and adolescents [23]. The instrument comprises 15 items related to the practice of physical activity in different domains (inside and outside of school and sedentary time) every day of the week. Each answer is scored on a 5-point Likert Scale ranging from 1 to 5. The questionnaire divided into three sections: (1) activity at school, (2) activity out of school, and (3) screen time. Physical activity was measured as the average of activity at school (i.e., as activity during physical education class, lunch, and recess) and out of school (i.e., activity before school, activity right after-school, activity during the evening, and activity on each weekend day). Screen time was calculated by inverting values from positive to negative according to the nature of the variable and calculating the average value of all responses related to screen media (i.e., watching television, playing video games, using the computer, and using a cell phone).

Sleep duration. Sleep duration was measured using the self-reported sleep questionnaire. This instrument is a valid and reliable measure to assess sleep duration among adolescents [25]. The questionnaire has four questions about usual week-day and weekend bedtimes and wake-up times. Daily sleep time was calculated by weighting weekdays and weekend days using a ratio of 5:2 (i.e., ([Daily sleep duration on weekdays ∗ 5] + [Daily sleep duration on weekend days ∗ 2]/7)).

Health-related quality of life. Participants were assessed using the Spanish version of the KIDSCREEN-10 questionnaire [26]. The KIDSCREEN-10 is a valid and reliable measure to assess health-related quality of life in youths [27]. The scale comprises 10 items assessing the subjective perception of health and well-being. Each statement is scored on a 5-point Likert scale ranging from 1 (never/not at all) to 5 (always/extremely). Scores were calculated for each dimension according to the methods described by the authors of the original scale using Rash analysis. Higher scores indicate better health-related quality of life; for more details, see [27]. Likewise, the reliability analysis showed acceptable reliability for the present sample (α = 0.71).

Covariates. Participants self-reported age, sex, socioeconomic status, and body mass index. Socioeconomic status was measured using the Family Affluence Scale-II (FAS II) [28]. A socioeconomic status score ranging from 0 to 9 was calculated based on the responses to four questions. The body mass index was calculated as weight in kilograms divided by height in meters squared (kg/m^2^).

### 2.3. Procedure

First, the research team contacted the principal and teacher of each school to obtain permission for data collection. In this regard, a face-to-face meeting was arranged with the director of the educational center and the objective of the study was explained. Likewise, an informed consent document was provided, which students had to bring signed by their legal guardians or parents to participate in the study. Subsequently, the data collection was carried out by the authors of the present work, who informed the students about the contents of the questionnaire. The research team explained to the students that their participation was voluntary and anonymous. Only those students who returned their written informed consent participated in this study. Likewise, all the doubts raised by the students during the administration of the questionnaire were asked. This research was approved by the Ethics Committee of the University of Extremadura (145/2019). 

### 2.4. Data Analysis

Descriptive statistics are presented as means and standard deviation or percentages (%). The normal distribution of the data was checked, which suggested the use of parametric statistics. Differences in the study variables between boys and girls were tested via Student’s t-test for continuous variable and chi-square test for categorical variables. 

Regression analyses were used to examine the associations between physical activity (PA), screen time (ST), sleep duration (SD), and health-related quality of life. First, the association between each movement behavior (PA, ST, or SD) and health-related quality of life was examined using simple linear regressions (Y = β_0_ + β_1_X_i_) + E_i_). Second, the association between all possible combinations of movement behavior (PA + SC or, PA + SD or, SC+SD or, PA + SC + SD ... etc.) with health-related quality of life was examined by performing various models’ multiple regressions (Y = β_0_ + β_1_X_i_ + β_2_X_2_ + β_3_X_3_) + E_i_), adding a new study variable at each step. Finally, the predictive value of each individual or combined behavior related to quality of life was examined, taking into account gender differences. In this sense, gender analyzes were carried out separately, since a significant interaction was shown with the variables of 24 h movement behaviors (*p* < 0.05) in relation to quality of life. All statistical analyzes were performed using SPSS Statistics 24.0 for Windows (SPSS Inc, Chicago, IL, USA). The level of significance was set at *p* < 0.05.

## 3. Results

Table 1 shows descriptive statistics and bivariate correlation of the variables studied. In this sense, the descriptive ones showed higher scores in all the study variables for the male gender compared to the female (all, *p* < 0.01), except for sleep duration (*p* > 0.05). 

The correlation analyses revealed a significant positive association between physical activity and quality of life (*p* < 0.01). However, sleep time was significantly negatively associated with screen time (*p* < 0.01).

### Gender Differences Regarding Movement Behavior for Predicting Quality of Life

Table 2 shows the association between individual and combined movement behaviors with quality of life in boys and girls. In this sense, the regression analyzes that were carried out (simple linear regression analysis (step 1) and multiple regression analysis (steps 2 and 3)) showed the impact of each behavior individually and in combination on the quality of life. The results obtained show how the beta value varied each time a new behavior was included in each step. More specifically, step one shows the regression weights of each of the variables that make up movement behavior in the prediction of quality of life, showing only significance in PA behavior. (*p* < 0.01). In step two, the different possible combinations were considered, evaluating how the beta values varied when only two behaviors were considered. Finally, in step three, the variation in the Beta values was evaluated according to the combination of the three behaviors, resulting in a significant value (*p* < 0.05) that can explain the impact on quality of life.

Similarly, the results revealed greater importance of physical activity behavior for boys compared to girls (*p* < 0.01). For its part, the female gender only showed significance in sleep for the regression model of the three behaviors combined (*p* < 0.50).

Subsequently, all regression models were repeated (simple linear regression analysis (step 1) and multiple regression analysis (steps 2 and 3)) but this time adjusting for the study covariates (sex, age, BMI, and socioeconomic status). This fact caused the results found to be stricter. In relation to this, it was observed that the behavior of physical activity together with time in front of the screen were the variables that best predicted the quality of life for the total sample (*p* < 0.05). Regarding gender differences, adjusted regression models showed that only men’s physical activity significantly predicts quality of life (*p* < 0.05). On the other hand, for the female gender, physical activity negatively affected the quality of life, although this prediction was not significant.

## 4. Discussion

The present study examined the individual and combined association between movement behaviors (physical activity, screen time, and sleep time) and quality of life in boys and girls. The main finding of this study is that a higher physical activity was positively associated with quality of life, especially in boys.

In this regard, the results found by Gonipath et al. [16] and Warner et al. [29] showed that changes in physical activity levels were related to positive changes in quality of life. These results could be explained by the benefits that the practice of physical activity represents in physical and psychosocial health, which is why the quality of life of adolescents is affected. Likewise, our results show that sleep time was positively and significantly related to quality of life. In this sense, previous studies showed a negative association between quality of sleep with a lesser quality of life and quality of life in children [14,30]. In this sense, lower quality of sleep was related to stress and depression [31,32], which could have a significant association with the quality of life of children and adolescents.

On the other hand, the regression models were presented without taking into account the impact of the covariates, and only the sex of the participants showed different findings. For unadjusted regression models, our findings showed that only individual physical activity behavior and combined sleep and screen time predict quality of life. In this sense, previous studies based on the "paradigm of the study of the behavior of segmented movement” [33] are in line with our results. Thus, several investigations highlight among their results the benefits of physical activity, showing that children and adolescents who maintain an active lifestyle exhibit better physical and psychosocial health (for example, quality of life or improvement in body composition) than those who do not [34,35,36]. In this regard, the work of Zurita-Ortega et al. [37] noted that when levels of physical activity increase, there is an overall increase in quality of life. Likewise, previous studies showed the relationship between quality of life, sleep time and screen time. In this regard, poor sleep time can have consequences on mood and the immune system [14,30,38], in addition to causing problems related to weight gain, which may be caused by increased use of television, video consoles, and mobile devices [39]. Likewise, in the unadjusted model for the whole sample, where the three behaviors (physical activity, screen time, and sleep time) were considered together, our findings showed that the sum of all of them predicted the quality of life; these results are similar to those found by Marques et al. [40] where healthy behaviors, such as physical activity, healthy eating, and sleep time, were strongly related to quality of life.

Considering the unadjusted regression model for children, our results show that only physical activity behavior had a significant predictive value for quality of life. This fact could be due to different leisure options, differences in motor skills, and gender stereotypes that sometimes limit the practice of physical activity [41,42]. However, for girls in the unadjusted regression models, our findings highlighted that only sleep time combined with physical activity and screen behaviors were important for quality of life. In this sense, there are differences in the results obtained by previous studies [20,43,44]. Furthermore, in contrast to our findings, the studies by Guimarães et al. [43] and Hesketh et al. [44] showed a higher adherence to movement behaviors. More specifically in line with our results, Guimarães states that girls show more ease in complying with sleep recommendations compared to their peers. A possible explanation for this fact is that not all healthy behaviors are equally important to predicting quality of life in the different genders. In this sense, it may happen that children with fewer hours of sleep are also those who consume more screen time due to the influence of new technologies (smartphones, game consoles, computers) that can cause difficulty sleeping, fatigue, and headache [14,45]; however, it could also happen that parenting style and home conditions influence their quality of life [46].

Finally, in the models adjusted for the covariates, our results showed significance for physical activity and screen time in the combination of the three behaviors in the total sample and in physical activity for males. In this sense, our findings are consistent with those found in previous studies [47,48,49] where the amount and intensity of physical activity are related to a better quality of life. Likewise, the fact that physical activity was only significant in predicting quality of life in boys could be due to leisure opportunities, differences in social and cultural roles, and psychological attributes [50,51]. On the other hand, individual or combined healthy behaviors did not show significance for the prediction of quality of life in this adolescent population.

The findings of the present study support the evidence that healthy behaviors should be evaluated on the whole, since not all healthy behaviors are equally important in the physical and psychological well-being of adolescents. Likewise, the following stand out as strengths of the present study: the novelty of the 24 h behavioral movement paradigm in the Spanish context, its contribution to addressing some of the gaps in the scientific literature [7,13], the use of a comprehensive set of covariates that provide more robustness to the results found, the finding that not all behaviors are equally important in the quality of life for the different genders, and finally the practical implications derived from this study. However the study has some limitations, such as its design, which does not allow establishing cause−effect relationships; the sample size, which does not allow the results to be extrapolated to other populations; the approach of considering the sample and subdividing it into participants that comply with the recommendations of the movement behaviors compared to those that do not; and the measurement instruments based on self-reports, which do not allow an exact assessment of the results but rather close to reality.

## 5. Conclusions

The present work concludes that healthy behaviors throughout the day are important for a better quality of life in adolescents. However, not all behaviors are equally important in the perception of the well-being of different genders. Therefore, the general conclusion is that the incidence of physical activity on quality of life is highlighted as the main behavior in predicting quality of life for a population of school adolescents. Furthermore, the need to promote these healthy habits from the educational context at an early age is highlighted. Future studies aimed at evaluating the impact of healthy behaviors should include eating habits as behavior in addition to physical activity, sleep time, and sedentary time to link them with different psychosocial benefits. Likewise, it is convenient to study the different sedentary behaviors separately (use of mobile phones, video game consoles, computers) to assess whether they all have a negative impact on the psychosocial well-being of children and adolescents.

### Practical Implications

The present work contributes among its implications a new perspective of promoting healthy behaviors to improve the well-being of children and adolescents. In this sense, future intervention work should focus on the promotion of healthy habits throughout the day, developing activities that promote the development of adequate sleep habits, avoiding sedentary leisure, and complying with the recommendations of moderate to vigorous physical activity among children and adolescents, thanks to the use of active teaching methods in primary and high schools. 

Promoting healthy behaviors in the school environment can contribute significantly to improving the well-being of children and adolescents. In this sense, health should be promoted from a global perspective within the educational context, using active methodologies and the development of projects that encourage greater involvement of students and their families, thus favoring the collaboration of teachers, students, and families in the development of activities that promote healthy habits, such as the correct use of new technologies, adequate sedentary habits, and compliance with the recommendations of moderate to vigorous physical activity in children and adolescents.

## Figures and Tables

**Table 1 ijerph-18-07550-t001:** Descriptive statistics and bivariate correlations of the variables studied.

Variables of Study	Total *n* = 319		Boys *n* = 163		Girls *n* = 156		*p*	Correlation			
	*M*	*DT*	*M*	*DT*	*M*	*DT*		1	2	3	4
1. Physical activity (1–5)	3.30	0.70	3.44	0.68	3.16	0.69	**0.00**	−	0.06	0.04	0.15 **
2. Screen Time	3.31	0.57	3.40	0.57	3.20	0.55	**0.00**	−	−	−0.18 **	0.10
3. Sleep duration (h/day)	8.08	0.96	8.88	0.96	8.75	1.23	0.24	−	−	−	0.11 *
4. Quality-of-life (1−5)	3.84	0.57	3.92	0.55	3.75	0.59	**0.00**	−	−	−	−

Notes: bold = significance; *p* * < **0.05**; *p* ** < **0.01**.

**Table 2 ijerph-18-07550-t002:** Associations between each of the three movement behaviors (physical activity, screen time, and sleep duration) and quality of life in the whole sample and by sex.

Models	All Sample				Boys				Girls			
	R^2^	β	*p*	CI 95%	R^2^	β	*p*	CI 95%	R^2^	β	*p*	CI 95%
Unadjusted												
Step 1												
PA	0.025	0.157	**0.005**	[0.028, 0.154]	0.098	0.313	**0.000**	[0.094, 0.263]	−0.005	−0.041	0.611	[−0.119, 0.070]
SC	0.010	−0.102	0.069	[−0.123, 0.005]	0.012	−0.111	0.158	[−0.147, 0.024]	0.002	−0.046	0.569	[−0.125, 0.069]
SD	0.012	0.109	0.055	[−0.001, 0.127]	0.003	0.057	0.476	[−0.059, 0.127]	0.013	0.138	0.090	[−0.012, 0.166]
Step 2												
PA + SC	0.033	0.151	**0.007**	[0.024, 0.151]	0.111	0.314	**0.000**	[0.095, 0.263]	0.004	−0.044	0.586	[−0.122, 0.069]
		−0.093	0.095	[−0.117, 0.009]		−0.114	0.130	[−0.144, 0.019]		−0.049	0.547	[−0.127, 0.068]
PA + SD	0.035	0.151	**0.007**	[0.024, 0.151]	0.098	0.309	**0.000**	[0.091, 0.262]	0.022	−0.050	0.541	[−0.126, 0.066]
		0.102	0.068	[−0.004, 0.123]		0.077	0.310	[−0.043, 0.135]		0.144	0.080	[−0.010, 0.170]
SC + SD	0.028	−0.131	**0.022**	[−0.141, −0.011]	0.019	−0.128	0.114	[−0.158, 0.017]	0.026	−0.086.	0.300	[−0.154, 0.048]
		0.133	**0.020**	[0.012, 0.142]		0.079	0.325	[−0.047, 0.141]		158	0.059	[−0.003, 0.180]
Step 3												
PA + SC + SD	0.048	0.142	**0.011**	[0.019, 0.146]	0.115	0.310	**0.000**	[0.092, 0.263]	0.030	−0.059	0.473	[−0.132, −0.061]
		−0.120	**0.035**	[−0.134, −0.005]		−131	0.089	[−0.156, 0.011]		−0.092	0.271	[−0.158, 0.045]
		0.125	**0.028**	[0.008, 0.137]		0.101	0.192	[−0.030, 0.150]		0.166	**0.050**	[0.000, 0.185]
Adjusted												
Step 1												
PA	0.082	0.114	**0.045**	[0.001, 0.130]	0.131	0.290	**0.000**	[0.078, 0.247]	0.060	−0.070	0.399	[−0.139, 0.056]
SC	0.080	−0.106	0.065	[−0.128, 0.002]	0.058	−0.108	0.179	[−0.148, 0.028]	0.066	−0.104	0.207	[−0.165, 0.036]
SD	0.071	0.059	0.325	[−0.034, 0.103]	0.047	0.058	0.483	[−0.062, 0.130]	0.058	0.043	0.639	[−0.078, 0.126]
Step 2												
PA + SC	0.092	0.112	**0.048**	[0.000, 0.128]	0.144	0.294	**0.000**	[0.081, 0.249]	0.072	−0.077	0.349	[−0.143, 0.051]
		−0.103	0.070	[−0.126, 0.005]		−0.119	0.124	[−0.150, 0.018]		−0.110	0.186	[−0.169, 0.033]
PA + SD	0.083	0.113	0.051	[0.000, 0.130]	0.128	0.286	**0.000**	[0.000, 0.004]	0.062	−0.065	0.440	[−0.138, 0.060]
		0.057	0.342	[−0.035, 0.101]		0.077	0.332	[−0.047, 0.138]		0.049	0.596	[−0.075, 0.130]
SC + SD	0.085	−0.124	**0.034**	[−0.140, −0.005]	0.062	−0.125	0.334	[−0.050, 0.145]	0.072	−0.123	0.146	[−0.180, 0.027]
		0.081	0.183	[−0.022, 0.116]		0.081	0.130	[−0.160, 0.021]		0.063	0.499	[−0.067, 0.138]
Step 3												
PA + SC + SD	0.096	0.109	0.058	[−0.002, 0.127]	0.145	0.291	**0.000**	[0.079, 0.250]	0.078	−0.077	0.362	[−0.144, 0.053]
		−0.121	0.039	[−0.138, −0.004]		−0.136	0.085	[−0.185, 0.011]		−0.130	0.126	[−0.185, 0.023]
		0.078	0.198	[−0.024, 0.127]		0.102	0.203	[−0.033, 0.154]		0.071	0.448	[−0.063, 0.143]

Note. The reported β values are standardized coefficients; Model adjusted by sex, age, socioeconomic status (EUR), and body mass index; PA: physical activity; SC: screen time; SD: sleep duration; bold = significance.

## Data Availability

The data presented in this study are available on request from the corresponding author.

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
