# Peer review of "Health Promotion through Movement Behaviors and Its Relationship with Quality of Life in Spanish High School Adolescents: A Predictive Study"

_ijerph, 2021, doi:10.3390/ijerph18147550_

Round 1
Reviewer 1 Report
Dear authors,
Thank you very much for the revision. I am still afraid I could follow your approach and problems may also arise for other readers as well.
Following points were not clear to me.
1) If you reversed the scale of screen time, this needs to be included in the method section as well.
2) In general, I think it is important to describe the regression approach very clearly in the method section. What does the variable PA+SC (or PA+SC+SD) mean, how was it calculated etc.?
3) Hence, I do not fully understand what the results e.g. in Step 3 mean. There are three different beta coefficients but still one variable (PA+SC+SD); same for step 2. Could you please explain how many (and which) variables were included in this step. How were they derived?
4) So, unfortunately, I still have strong doubts that the combination of different guidelines will work in your case. For the variables PA+SD, for instance, did you sum up those minutes and create a new variable? The problem is that the relationship between sleep duration and health is u shaped that is why guidelines do not recommend very low or high sleep times (only e.g. 9-11h). If you use simple summarised minutes, it will not account for the curvilinear relationship; it will just mean more minutes are better, always. This is why previous studies often created groups, e.g. meeting sleep guidelines vs not (below 9 or above 11h).
5) Again, I thought that the Canadian 24h movement guidelines recommend 9-11 hours for children aged 5-13y (not 8-10)? You may have a look here:
https://cdnsciencepub.com/doi/full/10.1139/apnm-2016-0151
6) Adjusted regression models. Which control variables were included in the separate models for boys and girls? You could make clear the distinction between the total model and the separate ones as there is no need to adjust for gender when running the model in each group.
7) Because you can still find the terms "independent, individual, combined" throughout the manuscript, readers may also get confused what you mean in your regression models. I suggest double-checking throughout the manuscript. for example, most people in epidemiological health research would think of indepedent associations when the variable (e.g. PA) is adjusted by another behavior (e.g., sleep), + perhaps adjustments for confounders.
8) Footnote Table 2. If only income was used you could call the variable income rather than socio-economic status (which is often derived in other ways).
Author Response
Cover letter
Dear reviewer, we appreciate all your suggestions for improvement, as they help to give more robustness and rigor to the study. In the following lines we will try to give answers to all your suggestions for improvement.
#Reviewer 1
Comments and suggestions
1) If you reversed the scale of screen time, this needs to be included in the method section as well.
2) In general, I think it is important to describe the regression approach very clearly in the method section. What does the variable PA+SC (or PA+SC+SD) mean, how was it calculated etc.?
3) Hence, I do not fully understand what the results e.g. in Step 3 mean. There are three different beta coefficients but still one variable (PA+SC+SD); same for step 2. Could you please explain how many (and which) variables were included in this step. How were they derived?
4) So, unfortunately, I still have strong doubts that the combination of different guidelines will work in your case. For the variables PA+SD, for instance, did you sum up those minutes and create a new variable? The problem is that the relationship between sleep duration and health is u shaped that is why guidelines do not recommend very low or high sleep times (only e.g. 9-11h). If you use simple summarised minutes, it will not account for the curvilinear relationship; it will just mean more minutes are better, always. This is why previous studies often created groups, e.g. meeting sleep guidelines vs not (below 9 or above 11h).
5) Again, I thought that the Canadian 24h movement guidelines recommend 9-11 hours for children aged 5-13y (not 8-10)? You may have a look here:
https://cdnsciencepub.com/doi/full/10.1139/apnm-2016-0151
6) Adjusted regression models. Which control variables were included in the separate models for boys and girls? You could make clear the distinction between the total model and the separate ones as there is no need to adjust for gender when running the model in each group.
7) Because you can still find the terms "independent, individual, combined" throughout the manuscript, readers may also get confused what you mean in your regression models. I suggest double-checking throughout the manuscript. for example, most people in epidemiological health research would think of indepedent associations when the variable (e.g., PA) is adjusted by another behavior (e.g., sleep), + perhaps adjustments for confounders.
8) Footnote Table 2. If only income was used you could call the variable income rather than socio-economic status (which is often derived in other ways).
Responses to the reviewer
Reviewer: 1) If you reversed the scale of screen time, this needs to be included in the method section as well.
Authors: Dear reviewer, we appreciate your suggestion. In this regard, the information in the manuscript has been modified, adding that the screen time scale has been inverted.
Pag 3, line 108-110 “Screen time was calculated by inverting values ​​from positive to negative according to the nature of the variable and calculating the average value of all responses related to screen media.”
Reviewer:2) In general, I think it is important to describe the regression approach very clearly in the method section. What does the variable PA+SC (or PA+SC+SD) mean, how was it calculated etc.?
Authors: We greatly appreciate your suggestion for improvement. In this sense, within the method section, specifically in the data analysis part, we have included an explanation of the regression approach accompanied by the simple and multiple linear regression equations. We have also provided some acronyms on Physical Activity (PA), Screen Time (ST), and Sleep Duration (SD) to make it easier for the reader.
Pag 4, line 151-157 “Regression analyses were used to examine the associations between physical activity (PA), screen time (ST), sleep duration (SD) and health-related quality of life. First, the association between each movement behavior (PA, ST, or SD) and health-related quality of life was examined using simple linear regressions (Y = β0 +β1Xi) + Ei). Second, the association between all possible combinations of movement behavior (PA + SC or PA + SD or PA + SC + SD ...) with health-related quality of life was examined by performing various models’ multiple regressions (Y = β0 +β1Xi + β2X2 + β3X3) + Ei), adding a new study variable at each step.”
Reviewer: 3) Hence, I do not fully understand what the results e.g. in Step 3 mean. There are three different beta coefficients but still one variable (PA+SC+SD); same for step 2. Could you please explain how many (and which) variables were included in this step. How were they derived?
Authors: We appreciate your question and that you give us the opportunity to better explain the results. In this sense, what is intended to show with the different steps (steps 1, 2 and 3 and beta coefficients is how the predicted quality of life varies each time we insert into the equation a new behavioral variable that explains the quality of life. Thus, when we talk about (PA + SC + SD) in step 3 we refer to the variation of beta values ​​when these 3 behaviors combined try to predict quality of life. In the same way, in step two the different possible combinations were considered, evaluating how the beta values ​​varied when only two behaviors of the movement behavior were considered.
Page 4-5, line 175-198 “Table 2 showed the association between independent and combined movement behaviors with quality of life in boys and girls. In this sense, the regression analyzes that were carried out (simple linear regression analysis "step 1", and multiple regression analysis "step 2 and 3") showed the impact of each behavior individually and in combination on the quality of life. The results obtained showed how the beta value varied each time a new behavior was included in each step. More specifically, step one showed the regression weights that the variables of movement behavior, individually, predict quality of life, showing only significance in PA behavior (p <0.01). In step two, the different possible combinations were considered, evaluating how the beta values ​​varied when only two behaviors were considered. Finally, in step three, the variation of the Beta values ​​was evaluated according to the combination of the three behaviors, resulting in a significant value (p <0.05) that can explain the impact on quality of life.
Similarly, the results revealed a greater importance in physical activity behavior for boys compared to girls (p < .01). For its part, the female gender only showed significance in sleep for the regression model of the three behaviors combined (p < .50).
Subsequently, all regression models were repeated (simple linear regression analysis "step 1", and multiple regression analysis "step 2 and 3"), but this time adjusting for the study covariates (sex, age, BMI, and socioeconomic status). This fact caused the results found to be stricter. In relation to this, it was observed that the behavior of physical activity together with time in front of the screen were the variables that best predicted the quality of life for the total sample (p <.05). Regarding gender differences, adjusted regression models showed that only men's physical activity significantly predicts quality of life (p <.05). On the other hand, for the female gender, physical activity negatively affected the quality of life, although this prediction was not significant.”
Reviewer: 4) So, unfortunately, I still have strong doubts that the combination of different guidelines will work in your case. For the variables PA+SD, for instance, did you sum up those minutes and create a new variable? The problem is that the relationship between sleep duration and health is u shaped that is why guidelines do not recommend very low or high sleep times (only e.g. 9-11h). If you use simple summarised minutes, it will not account for the curvilinear relationship; it will just mean more minutes are better, always. This is why previous studies often created groups, e.g. meeting sleep guidelines vs not (below 9 or above 11h).
Authors: The authors greatly appreciate your question and suggestion, in addition, we understand your concern about this issue, since, as you well expressed, there is a curvilinear relationship between physical activity and the duration of sleep-in terms of the benefits derived from it. In this regard, the authors assume that our study has this limitation. However, the object of our study has not been to evaluate differences in the quality of life of the population that complies with the 24-hour behavioral recommendations versus those that do not comply with the recommendations, but rather to evaluate the impact of each individual behavior and combined in quality of life and analyze gender differences.
In this sense, thanks to your recommendation, we have added your suggestion as a limitation of the study.
Pag 8, 272-274 “the approach of considering the sample and subdividing it into participants that comply with the recommendations of the movement behaviors compared to those that do not”
Reviewer: 5) Again, I thought that the Canadian 24h movement guidelines recommend 9-11 hours for children aged 5-13y (not 8-10)? You may have a look here:
https://cdnsciencepub.com/doi/full/10.1139/apnm-2016-0151
Authors: We greatly appreciate your suggestion and apologize for any oversight regarding our sleep recommendations. In this sense, we have modified the dream hours recommendations according to the information you have provided us.
Pag 2, line 69 “9 to 11”
Reviewer: 6) Adjusted regression models. Which control variables were included in the separate models for boys and girls? You could make clear the distinction between the total model and the separate ones as there is no need to adjust for gender when running the model in each group.
Authors: Thank you very much again for giving us the opportunity to explain this matter. In this regard, we would like to say that when we talk about fitted regression models, we mean that they fit the covariates indicated in the method (measurement) section. These variables refer to sex, age, BMI, and socioeconomic status. However, when the models were segmented for sex in the regressions, they only adjusted for the covariates of age, BMI, and socioeconomic status. Likewise, we must add that in the adjusted general model, gender was considered in addition to the covariates mentioned above. On the contrary, in the unadjusted models the covariates were not considered, only the models were segmented according to sex.
Reviewer: 7) Because you can still find the terms "independent, individual, combined" throughout the manuscript, readers may also get confused what you mean in your regression models. I suggest double-checking throughout the manuscript. for example, most people in epidemiological health research would think of indepedent associations when the variable (e.g., PA) is adjusted by another behavior (e.g., sleep), + perhaps adjustments for confounders.
Authors: We appreciate your suggestion. In this regard, in order not to confuse readers from different research areas, and considering the argument that you have given us, we have decided to review the entire manuscript and unify the terms of the entire manuscript in relation to the term "individual", since Articles such as the one you have provided in relation to movement behavior guidelines use the term "individual behaviors" or "individual movement behaviors" of movement behavior. Also, I add some references that attest to the use of this term:
https://cdnsciencepub.com/doi/full/10.1139/apnm-2016-0151
Sampasa-Kanyinga, H., Colman, I., Goldfield, G. S., Janssen, I., Wang, J., Hamilton, H. A., & Chaput, J. P. (2021). 24-h Movement Guidelines and Substance Use among Adolescents: A School-Based Cross-Sectional Study. International journal of environmental research and public health, 18(6), 3309.
Okely, A. D., Tremblay, M. S., Reilly, J. J., Draper, C. E., & Bull, F. (2018). Physical activity, sedentary behaviour, and sleep: movement behaviours in early life. The Lancet Child & Adolescent Health, 2(4), 233-235.
Reviewer: 8) Footnote Table 2. If only income was used, you could call the variable income rather than socio-economic status (which is often derived in other ways).
Authors: We welcome your suggestion. However, the authors have used the FAS scale (THE FAMILY AFFLUENCE SCALE) for the evaluation of the socioeconomic level, since asking about the income level can cause rejection in the sample when responding. In this regard, this scale estimates the family's socioeconomic level through 4 items related to the family's additive level (number of cars, single or shared room, vacation trips throughout the year, and number of computer devices in the home).

Reviewer 2 Report
Overall revision is fine. The only minor changes are recommended.
Line 15: Used “vigorous” physical activity but Line 32 used “high” physical activity. Need to clarify the differences between two terms for reader understanding.
Line 52: Results of the study shouldn’t be mentioned in the introduction section
Author Response
Cover letter
Dear reviewer, we appreciate the positive assessment you have made of the manuscript and the suggested improvement suggestions. In this regard, in the following lines we will try to answer all your suggestions.
#Reviewer 2
Overall revision is fine. The only minor changes are recommended.
Line 15: Used “vigorous” physical activity but Line 32 used “high” physical activity. Need to clarify the differences between two terms for reader understanding.
Line 52: Results of the study shouldn’t be mentioned in the introduction section.
Responses to the reviewer
Reviewer: Line 15: Used “vigorous” physical activity but Line 32 used “high” physical activity. Need to clarify the differences between two terms for reader understanding.
Authors: Dear reviewer, we appreciate your suggestion and apologize for the mistake we made. In this sense, the term vigorous refers to the intensity of the PA, and the term "high" refers to the amount of PA that a person performs throughout the day. In this regard, when we talk about recommendations, we refer to the accumulation of PA minutes as suggested by the WHO or the Canadian Guide to Movement Behavior. For all this, we have unified both terms with the term "high."
Page 1, line 16 “High”
Reviewer: Line 52: Results of the study shouldn’t be mentioned in the introduction section.
Authors: We appreciate your suggestion for improvement. However, when the authors mention certain studies on page 2, starting from line 50 (study 13, and 20), we are describing the results of these two studies, and arguing the reasons that led us to carry them out. of the present manuscript.
another modification
Finally, it should be noted that the authors have added some more lines to the conclusion and strengths of the study that we believe give the manuscript more quality.
Page 8, line 269, 280-281.

Reviewer 3 Report
The article has been significantly improved compared to the original version and is now complete in my opinion. The aim of the study was to analyze the relationship between health-related quality of life and individuals and combinations of motor behavior, and to identify gender differences in this relationship. The research procedure is understandable and repeatable. The results are described in clear and understandable language. The abbreviations below the table are explained. The work is written correctly in accordance with the stages of the scientific method. The methodological description has been extended. The discussion and conclusions remained unchanged. Conclusions answer the research question and result from the conducted analysis. I suggested a paragraph structure and a clear division into theoretical and application conclusions, but the current layout is also clear. References have been verified and limited.
Author Response
Cover letter
# Reviewer 3
The article has been significantly improved compared to the original version and is now complete in my opinion. The aim of the study was to analyze the relationship between health-related quality of life and individuals and combinations of motor behaviors, and to identify gender differences in this relationship. The investigation procedure is understandable and repeatable. The results are described in clear and understandable language. Abbreviations are explained below the table. The work is correctly written according to the stages of the scientific method. The methodological description has been expanded. The discussion and conclusions were unchanged. The conclusions respond to the research question and are the result of the analysis carried out. I suggested a clear paragraph structure and division into theoretical and application conclusions, but the current design is also clear. References have been verified and limited.
Authors: Dear reviewer, we appreciate your positive evaluation of all the modifications carried out to improve the quality of the manuscript. In this sense, the authors appreciate the feedback you have given us, recognizing the improvement work that has been carried out.

Reviewer 4 Report
- The abstract of the article is well prepared. It has the methodology identified and indicates the direction of the results obtained by the authors in this research.
- [line 90-91] The authors have clearly identified which criteria were selected, for the definition of the sample, which is very important ("... the sample was carried out through an intentional sampling for convenience ...").
- [line 97] The authors mention that they used the “Youth Activity Profile Questionnaire - Spain (YAP-S)”, but was the questionnaire designed by whom? The authors also mention that the assessment is made from 1 to 5, is the Likert Scale used? This information should be clarified in the article.
- [line 122-123] The authors say: “Likewise, it showed an acceptable reliability, for the present sample (α = .71).”, but how was it validated by the authors? What statistical test did the authors perform? Alpha Cronbach? This information should be provided to the scientific community.
- Which program or software did the authors use, to do the data analysis and statistical analysis?
- The authors should review the formatting of the text, because it seems that the paragraphs are too far to the left of the page.
Author Response
Cover letter
Dear reviewer, we appreciate your suggestions for improvement as they provide the manuscript with greater quality and robustness. In this regard, the following lines will try to respond to all your suggested improvement suggestions.
#Reviewer 4
The abstract of the article is well prepared. It has the methodology identified and indicates the direction of the results obtained by the authors in this research.
[line 90-91] The authors have clearly identified which criteria were selected, for the definition of the sample, which is very important ("... the sample was carried out through an intentional sampling for convenience ...").
[line 97] The authors mention that they used the “Youth Activity Profile Questionnaire - Spain (YAP-S)”, but was the questionnaire designed by whom? The authors also mention that the assessment is made from 1 to 5, is the Likert Scale used? This information should be clarified in the article.
[line 122-123] The authors say: “Likewise, it showed an acceptable reliability, for the present sample (α = .71).”, but how was it validated by the authors? What statistical test did the authors perform? Alpha Cronbach? This information should be provided to the scientific community.
Which program or software did the authors use, to do the data analysis and statistical analysis?
The authors should review the formatting of the text because it seems that the paragraphs are too far to the left of the page.
Responses to the reviewer
Reviewer: [line 90-91] The authors have clearly identified which criteria were selected, for the definition of the sample, which is very important ("... the sample was carried out through an intentional sampling for convenience ...").
Authors: We appreciate the feedback given on the improvement that the authors have made on the sample selection criteria. In this regard, the full text reads as follows:
“The selection of the sample was carried out through an intentional sampling for convenience according to the distance of the schools to the research staff in charge of data collection, the willingness to collaborate on the part of the teaching staff, and the time required for the researcher to travel towards collaborating centers. Likewise, all subjects consented to their participation in the study.”
Reviewer: [line 97] The authors mention that they used the “Youth Activity Profile Questionnaire - Spain (YAP-S)”, but was the questionnaire designed by whom? The authors also mention that the assessment is made from 1 to 5, is the Likert Scale used? This information should be clarified in the article.
Authors: Thank you very much for your suggestion. In this regard, in the description of the instrument, in line 99, we mention with the citation 24 who designed the instrument. In this regard, we have decided to add the name of the author to the text (Saint-Maurice and Welk [24]). On the other hand, regarding the type of scale we have added the term Likert scale in line 103.
Reviewer: [line 122-123] The authors say: “Likewise, it showed an acceptable reliability, for the present sample (α = .71).”, but how was it validated by the authors? What statistical test did the authors perform? Alpha Cronbach? This information should be provided to the scientific community.
Authors: Thank you very much for your suggestion. In this regard, the authors performed a reliability analysis for this instrument in the present sample, and as stated in the text, we obtained a Cronbach's alpha of .71. In this sense we have added new information on page 3, line 126.
Page 3, line 126 “Likewise, the reliability analysis it showed an acceptable reliability, for the present sample (α = .71).”
Reviewer: Which program or software did the authors use, to do the data analysis and statistical analysis?
Authors: Thanks for your question. In this regard, the authors in the data analysis section, on line 162 we indicate the statistical package used.
Page 4, line 162 “All statistical analyzes were performed using SPSS Statistics 24.0 for Windows (SPSS Inc, Chicago, IL). The level of significance was set at p< .05.”
Reviewer: The authors should review the formatting of the text because it seems that the paragraphs are too far to the left of the page.
Authors: Dear reviewer, the authors are aware of this aspect of the format, but we believe that this is due to the specific format that the current scientific journal maintains.

This manuscript is a resubmission of an earlier submission. The following is a list of the peer review reports and author responses from that submission.
Round 1
Reviewer 1 Report
Overall
Dear all,
Thank you for inviting me to review this manuscript.
The study investigates whether different 24h human movement behaviors relate to quality of life in adolescents. Understanding how daily behaviors affect our health is important. This was done by several studies in the past but adequately addressing the collinearity between those behaviors is relatively new in (physical activity) research. I think the current study is important and provides valuable insights, also for the 24h movement guidelines.
However, I have strong concerns regarding the analytical approach and the quality how the results are presented (e.g., missing footnotes, no confidence intervals in the entire manuscript, incomplete description of the approach in the method section). In my opinion, two statistical problems needs to be addressed (see also my last comment before the discussion). First, the study aims to investigate differences in the association between boys and girls, but no formal interaction test was performed. Secondly, if I understood it correctly, minutes in different behaviors were summed up (e.g., daily minutes physical activity + screen time). Although this can be seen as an approach to combine these different behaviors, I doubt that we can interpret the regression coefficients in the way we want to. Other authors have used compositional data analysis, or a combination based on ‘healthy’ categories, which could then be appropriate (e.g., meeting all three recommendations vs not meeting them). I am happy to discuss the rationale for the present approach. Thank you very much.
Specific comments
Title
The information about health promotion in the title is a bit misleading as this is basic research trying to understand the relationship between movement behaviors and health. Perhaps revise it?
Introduction
P. 2., line 40. You may add information on the codependence of the behaviors. Why is this the case? Why could it be a problem when analyzing it (e.g., wider standard error, fluctuating regression coefficients)? I think this could be highlighted as studying the combinations is the aim of the present study (You may find some information here: https://pubmed.ncbi.nlm.nih.gov/27306435/; https://ijbnpa.biomedcentral.com/articles/10.1186/s12966-018-0707-z)
P. 2, line 50. The original study (ref 19) did not test for an interaction between sex and meeting the guidelines. I think stating that the effect was only observed in girls but not in boys has to be done carefully, as the effect estimates for boys and girls are pointing towards the same direction. I would rather include interaction terms first to explicitly test for differences in slopes between boys and girls (like a subgroup analysis after testing for formal interaction in RCTs).
P. 2, line 55. What are the implications of studying combinations of different 24h behaviors? Why is that important? The introduction could be strengthened by adding this information rather than just saying ‘only few studies investigated it, thus it is important to do it’.
P. 2, line 67. Could you please explain to me why no hypotheses regarding the behavior-health relationship were formulated, as several studies showed that these behaviors are related to several health outcomes, including mental health and quality of life? For example, I would expect the combination (e.g., meeting both sleep and activity guideline) would relate to higher quality of life than meeting none of them?
I think it might be worth stating and explaining the Canadian 24h movement guidelines somewhere in the introduction.
Methods
P. 2, line 80. Does this questionnaire measure total physical activity (including light, moderate, vigorous) or only MVPA in different settings? You may add this information for other readers. Also, I thought the questionnaire assesses other sedentary habits, in addition to screen time? Did you include this information in your analysis?
Related to that, the guidelines apply to different types of sedentary behaviors, so perhaps it might be preferred to include all sedentary information from the YAP-C questionnaire and call it total sedentary time (which should be limited to 2 hours a day)?
p. 2, line 80: Please provide further information on the validity and reliability of the YAP-S in your population (e.g., coefficients for reliability and validity). Often the authors of the validation study conclude that the validity is acceptable, but this does often not the case (e.g., correlation coefficient of 0.2). You may consider a more detailed description for the other self-reported measures as well.
P. 3, line 115. Could you provide more information on how many students were targeted/invited, and how many of them signed the form (consented), perhaps under section participants? Also, did you exclude some students due to any other reasons (e.g. outliers, missing data)? You could also mention the exact inclusion criteria for your study population.
P. 3, line 121. You may use a different wording here (‘verified’). Verifying is never possible, unfortunately.
P. 3, line 125. Perhaps you could use another word for ‘independent’ in the manuscript. Many researchers will think of adjusting for the other behaviors in a combined regression model (e.g., a model with physical activity but adjusted for one or two other behaviors). If I understood it right, you run separate bivariate regression models. It may be worth clarifying this here and perhaps call it ‘to assess the individual influence of each behavior on quality of life’? So individual and combined vs independent and combined? Just a suggestion.
P. 3, line 128. Are the results from the ANOVA in the main file? I am afraid I could not find it. Anyway, further ANOVAs may not be needed as the research question is already addressed/answered with linear regression models.
P. 3., section 2.4. You may add information about the covariates and which of them you have used for which model. For example, in the separate models for sex and boys there is no need to adjust for sex.
Results
Table 2. Please also provide 95% CI for all estimates as they provide important information about the uncertainty.
Table 2. Did I understand it correctly that you performed separate regression models for all those variables listed in the first column and that the last models were always adjusted for age, status etc. or did you mutually adjust the last model for all listed variables (e.g., physical activity adjusted by screen time and so on)?
Table 1 and 2. Please provide proper footnotes and the units of measurement for all variables (e.g., screen time (min/d)). In the notes you could also mention what the confounders were in the adjusted analyses.
Table 1: What does ‘DT’ mean? Perhaps SD is more often used?
After reading the result section, I have two doubts about the statistical approach in your study. First, the rationale is to study combined influences of 24h human movement behaviors on quality of life (because these behaviors strongly interrelate due to their closed structure and collinearity). So, some form of combination is necessary (e.g., others have used compositional data analysis or partial least squares regression, https://pubmed.ncbi.nlm.nih.gov/27306435/). For me it is unclear how this combination was done in the present study (e.g., variable ‘physical activity + sleep duration’). Is that the sum of minutes in activity and sleep or is it a variable representing meeting both recommendations vs none (or vs only meeting one), like this was done in previous studies https://pubmed.ncbi.nlm.nih.gov/27306434/ ? This was not well described in the method section. If it is a sum of minutes, it will not allow a valid estimate of how a ‘healthy combination’ of the two behaviors relate to more quality of life (pointing in the same direction!). In this example (activity+sleep), a linear (assumed positive) association with quality of life is assumed (higher minutes assume either healthier behavior; or non-healthier behavior, in general), but this does not address the guidelines (e.g., recommendations of 9-11 or 8-10 hours a day). What happens when someone sleeps 13 hours a day, the person still gets a higher score in the exposure variable but this non-linear relationship between sleep and health is not addressed (such high values would actually mean that someone shows a poor sleeping pattern). So, the relationship, even though continuous, does not always point to the same direction. Another example for screen time + physical activity. Assuming a sum of minutes, the coefficient of 0.10 does not tell us what we want, namely ‘more healthy minutes in those two behaviors (higher activity, less screen time) relates to more quality of life’, because the variables are reversed. A teenager with a value of 10 hours can do 8 hours of screen time and 2 hours of activity, while a teenager with 11 hours (which is now a higher value in the regression line; and we assume that this is uniformly better/worse) can do 9 hours of screen time / 2 hours of activity (worse) or 7 hours of screen time / 4 hours of activity (better). So, even though the values are increasing from 10 to 11 hours, we cannot be sure that the direction is the same. Hence, in my opinion the coefficients are not interpretable. Perhaps a different statistical approach (e.g., using combinations of ‘meeting two or more guidelines vs not’ instead of summarized minutes; compositional data analysis) should be implemented. However, I am happy to discuss the rationale of using summarized minutes in the present study. Secondly, a very brief point, I think testing whether the effect is different for boys and girls requires the inclusion of interaction terms in the main regression models. This will test whether the slopes are different rather than just showing separate results for boys and girls (one is then tempted to look at the p values and compare them, which should not be done).
Discussion and Conclusion
It could be important to highlight potential measurement problems for the main exposure variables. The collected information is based on self-reports, and you could acknowledge some problems with it (e.g., reporting errors when summing up information from the past 7 days, social desirability, problems in recalling lower intensity activities, telescoping, e.g. https://pubmed.ncbi.nlm.nih.gov/33536193/).
Related to that, you could also discuss how these methodological issues may relate to problems in the estimation of effects. For example, in Table 2, the beta estimates for screen time and physical activity are very similar (0.10, 0.11) but show p values just above or below the cut off (‘dancing around the threshold’). Labeling the one as an important influence and the other as not is tricky. So, you could discuss how any shortcomings of the study (e.g., measurement, representativeness of the sample, residual confounding due to missing information when including only one movement behavior in the model) could have impacted the results (e.g., including the width of the confidence intervals, reflecting some uncertainty in the effect estimates). Mentioning/discussing these issues is important, especially concerning the ‘non-significant’ results (e.g., you highlight that the main finding was the estimate of 0.11 for physical activity, line 157).
p. 4, line 154. As mentioned above, you may consider using the wording individual and combined distributions rather than ‘independent’. Independent could also mean adjusted for the others in the regression model.
p. 5, line 163. What are the differences between quality of sleep and sleep time, since your study addressed the time but here you refer to previous studies showing similar results concerning sleep quality?
The conclusion could be slightly revised so that it aligns with the main results of the study. In my opinion, the conclusion should rather stick to the main findings. For example, much information about promotion and eating behavior is provided but this was not addressed by the present study.
Author Response
Dear editor,
First of all, we want to thank the reviews made by the reviewers, as it improves the quality of this article. In this sense, we believe that as a result of these modifications our article is more robust. Below you will find the changes made according to the recommendations of each reviewer.
Reviewer #1
Major revision
Dear all,
Thank you for inviting me to review this manuscript.
The study investigates whether different 24h human movement behaviors relate to quality of life in adolescents. Understanding how daily behaviors affect our health is important. This was done by several studies in the past but adequately addressing the collinearity between those behaviors is relatively new in (physical activity) research. I think the current study is important and provides valuable insights, also for the 24h movement guidelines.
However, I have strong concerns regarding the analytical approach and the quality how the results are presented (e.g., missing footnotes, no confidence intervals in the entire manuscript, incomplete description of the approach in the method section). In my opinion, two statistical problems needs to be addressed (see also my last comment before the discussion). First, the study aims to investigate differences in the association between boys and girls, but no formal interaction test was performed. Secondly, if I understood it correctly, minutes in different behaviors were summed up (e.g., daily minutes physical activity + screen time). Although this can be seen as an approach to combine these different behaviors, I doubt that we can interpret the regression coefficients in the way we want to. Other authors have used compositional data analysis, or a combination based on ‘healthy’ categories, which could then be appropriate (e.g., meeting all three recommendations vs not meeting them). I am happy to discuss the rationale for the present approach. Thank you very much.
Specific comments
Title
Contribution of Reviewer. The information about health promotion in the title is a bit misleading as this is basic research trying to understand the relationship between movement behaviors and health. Perhaps revise it?
Answer: Dear reviewer, thanks for your comment. In this regard, we have changed the title. We hope that the title that is written below is more appropriate
Page 1, line 1-2 “Health promotion through movement behaviors and its relationship with quality of life in Spanish adolescents: a predictive study”
Introduction
Contribution of Reviewer. P. 2., line 40. You may add information on the codependence of the behaviors. Why is this the case? Why could it be a problem when analyzing it (e.g., wider standard error, fluctuating regression coefficients)? I think this could be highlighted as studying the combinations is the aim of the present study (You may find some information here: https://pubmed.ncbi.nlm.nih.gov/27306435/; https://ijbnpa.biomedcentral.com/articles/10.1186/s12966-018-0707-z)
Answer: Thank you for your contribution, which helps to better understand the manuscript. In this sense, we believe that the codependency of movement behaviors should be jointly and continuously valued since, as Tremblay et al (2016) introduce in Canadian 24-Hour Movement Guidelines for Children and Youth: movement behaviors should be evaluated from a holistic perspective in the 24-hour period. In this sense, these authors established these guidelines based on scientific evidence, where the fourth systematic review of this article, which evaluates the impact of physical activity, sleep and sedentary time, has an impact associated with important health indicators. Likewise, the cross-sectional study carried out by Carson et al. (2016) highlights the relevance of movement behaviors 24 hours a day for important health indicators.
https://cdnsciencepub.com/doi/full/10.1139/apnm-2016-0151
Therefore, following the reviewer's suggestions we have add the next sentence:
Page 2, line 43-45: “In this regard, it is important to evaluate movement behaviors from a holistic perspective in the 24-hour period, since these have an important impact on health indicators. Previous studies have linked quality of life with a higher level of physical activity…”
Contribution of Reviewer. P. 2, line 50. The original study (ref 19) did not test for an interaction between sex and meeting the guidelines. I think stating that the effect was only observed in girls but not in boys has to be done carefully, as the effect estimates for boys and girls are pointing towards the same direction. I would rather include interaction terms first to explicitly test for differences in slopes between boys and girls (like a subgroup analysis after testing for formal interaction in RCTs).
Answer: Thank you very much for the recommendation. In this regard, what the authors want to highlight with this reference [19] is the need for more studies to evaluate differences between boys and girls who meet the recommendations for movement behavior related to quality of life. Therefore, following their recommendations, the text has been modified, remaining as follows:
Page 2, line 52-57: “Despite these findings, mixed results have been reported between boys and girls in this relationship[19]. Specifically, the relationship between all three movement behaviours and health-related quality of life was significant only for interaction of the three behaviors in girls, whereas the interaction of the three behaviors in boys points in the same direction, but compared to the girls this was not significant. In addition, a greater importance was observed in relation to quality of life for those participants who especially complied with the recommendations for sleep and sedentary time [19]”
Contribution of Reviewer.P. 2, line 55. What are the implications of studying combinations of different 24h behaviors? Why is that important? The introduction could be strengthened by adding this information rather than just saying ‘only few studies investigated it, thus it is important to do it’.
Answer: We appreciate your recommendation. In this regard we have added the next information:
Pag 2, line 57-63 “Given that there is no convincing evidence, more studies that examine the relationship combinations of movement behaviors with health-related quality of life are needed [13,19]. In this sense, 24-hour movement behaviors represent a new paradigm under the concept "all day matters" Tremblay et al., (2016). This new paradigm, all supported by the scientific literature alludes to the importance of assessing movement behaviors 24 hours a day, compared to the paradigm that uniquely valued movement behavior related to health indicators (e.g. Physical activity) (Carson et al., 2016).”
References:
Carson, V., Tremblay, M. S., Chaput, J. P., & Chastin, S. F. M. (2016). Associations between sleep, sedentary time, physical activity and health indicators among Canadian children and youth using compositional analyses. Appl. Physiol. Nutr. Metab, 41.
Contribution of Reviewer. P. 2, line 67. Could you please explain to me why no hypotheses regarding the behavior-health relationship were formulated, as several studies showed that these behaviors are related to several health outcomes, including mental health and quality of life? For example, I would expect the combination (e.g., meeting both sleep and activity guideline) would relate to higher quality of life than meeting none of them?
I think it might be worth stating and explaining the Canadian 24h movement guidelines somewhere in the introduction.
Answer: We appreciate your recommendation. However, the authors chose not to state hypotheses due to little knowledge about the possible directionality of the results, since as can be seen throughout the manuscript, only for the boys who meet the physical activity recommendations, adjusted for the covariates, predict quality of life. On the other hand, we have followed your recommendation to add information about the Canadian 24 hour movement behavior page 2, line 63-67
Methods
Contribution of Reviewer. P. 2, line 80. Does this questionnaire measure total physical activity (including light, moderate, vigorous) or only MVPA in different settings? You may add this information for other readers. Also, I thought the questionnaire assesses other sedentary habits, in addition to screen time? Did you include this information in your analysis?
Related to that, the guidelines apply to different types of sedentary behaviors, so perhaps it might be preferred to include all sedentary information from the YAP-C questionnaire and call it total sedentary time (which should be limited to 2 hours a day)?
Answer: Thanks for your appreciation. In this regard, the authors have added the following information: The instrument comprises 15 items related to the practice of physical activity in different domains (inside and outside of school, and sedentary time) every day of the week (page 3, line 90-91). Also, in relation to the question of taking into account the different sedentary habits in the analysis. The authors report that they have been taken into account to calculate the total variable of sedentary time. This variable was formed by the mean value of all the responses related to screen time.
Finally, we would like to justify the reason why we named it screen time total, instead of total sedentary time. Thus, screen time was used because the instrument items include the time spent watching television (TV), playing video games, using the computer, using a cell phone, and a sedentary time element in general.
Contribution of Reviewer p. 2, line 80: Please provide further information on the validity and reliability of the YAP-S in your population (e.g., coefficients for reliability and validity). Often the authors of the validation study conclude that the validity is acceptable, but this does often not the case (e.g., correlation coefficient of 0.2). You may consider a more detailed description for the other self-reported measures as well.
Answer: Thanks for your recommendation. In this sense, the original manuscript showed a reliability of 0.52 to 0.79 and the ICC varied from 0.79 to 087.
Contribution of Reviewer p P. 3, line 115. Could you provide more information on how many students were targeted/invited, and how many of them signed the form (consented), perhaps under section participants? Also, did you exclude some students due to any other reasons (e.g. outliers, missing data)? You could also mention the exact inclusion criteria for your study population.
Answer: We welcome your suggestion for improvement.In this sense, the authors of the “procedure” section explained how the students participated.In this sense, only the students who did not return informed consent did not participate in the study.Likewise, of all the participants who participated, none were excluded. In addition, the following information about the sample selection process has been added
Page 3, line 85-88 “The selection of the sample was carried out through an intentional sampling for convenience according to the distance of the schools to the research staff in charge of data collection, the willingness to collaborate on the part of the teaching staff, and the time required for the researcher to travel towards collaborating centers. Likewise, all subjects consented to their participation in the study”
Contribution of Reviewer P. 3, line 121. You may use a different wording here (‘verified’). Verifying is never possible, unfortunately.
Answer: Thanks for your information. We have replaced verified with checked (page 3, line 130)
Contribution of Reviewer P. 3, line 125. Perhaps you could use another word for ‘independent’ in the manuscript. Many researchers will think of adjusting for the other behaviors in a combined regression model (e.g., a model with physical activity but adjusted for one or two other behaviors). If I understood it right, you run separate bivariate regression models. It may be worth clarifying this here and perhaps call it ‘to assess the individual influence of each behavior on quality of life’? So individual and combined vs independent and combined? Just a suggestion.
Answer: We appreciate your suggestion for improvement, as it contributes to a better understanding of the text. In this regard, the authors think that the individual word in relation to the combination of behaviors is more correct. For this reason the word independent has been modified. (Page, 3, line 134)
Contribution of Reviewer P. 3, line 128. Are the results from the ANOVA in the main file? I am afraid I could not find it. Anyway, further ANOVAs may not be needed as the research question is already addressed/answered with linear regression models.
Answer: Thank you very much for your observation. In this regard, the authors apologize as we were wrong with the concept of ANOVA. We have modified the phrase page 3 line 136-137: “Finally, the predictive value of each individual or combined behavior related to quality of life was examined.”
Contribution of Reviewer P. 3., section 2.4. You may add information about the covariates and which of them you have used for which model. For example, in the separate models for sex and boys there is no need to adjust for sex.
Answer: Thanks for your recommendation. In this regard, the authors describe the covariates in the measurement section. Likewise, in the adjusted models separated by sex, only the rest of the covariates have been taken into account.
Results
Contribution of Reviewer Table 2. Please also provide 95% CI for all estimates as they provide important information about the uncertainty.
Answer: Thanks for your recommendation. In this regard, the IC has been added. However, we have had to move the location of the table in the text due to the modifications in its content.
Localization: end of manuscript
Contribution of Reviewer Table 2. Did I understand it correctly that you performed separate regression models for all those variables listed in the first column and that the last models were always adjusted for age, status etc. or did you mutually adjust the last model for all listed variables (e.g., physical activity adjusted by screen time and so on)?
Answer: We, the authors, appreciate your question, as it gives us the opportunity to better explain the performance of the regression models. Thus, the procedure followed has been the realization of several regression models separately for all the variables of the first column, whose second part (adjusted) that refers to the entire sample was adjusted for sex, age, BMI, and socioeconomic status. The following columns are segmented by the sex of the participants, so the adjusted part refers to covariates such as age, BMI, and socioeconomic status.
Finally, it should be pointed out that each step of the model was introduced variables of the movement behavior, taking into account the covariates for the adjusted models.
Contribution of Reviewer Table 1 and 2. Please provide proper footnotes and the units of measurement for all variables (e.g., screen time (min/d)). In the notes you could also mention what the confounders were in the adjusted analyses.
Answer: We welcome your suggestion and have added explanatory footnotes on units of measure. However, the only unit that required units of measurement was sleep duration (h / day).
Contribution of Reviewer Table 1: What does ‘DT’ mean? Perhaps SD is more often used?
Answer: We appreciate your suggestion for improvement. In this sense, it has already been modified in the text.
Contribution of Reviewer After reading the result section, I have two doubts about the statistical approach in your study. First, the rationale is to study combined influences of 24h human movement behaviors on quality of life (because these behaviors strongly interrelate due to their closed structure and collinearity). So, some form of combination is necessary (e.g., others have used compositional data analysis or partial least squares regression, https://pubmed.ncbi.nlm.nih.gov/27306435/). For me it is unclear how this combination was done in the present study (e.g., variable ‘physical activity + sleep duration’). Is that the sum of minutes in activity and sleep or is it a variable representing meeting both recommendations vs none (or vs only meeting one), like this was done in previous studies https://pubmed.ncbi.nlm.nih.gov/27306434/ ? This was not well described in the method section. If it is a sum of minutes, it will not allow a valid estimate of how a ‘healthy combination’ of the two behaviors relate to more quality of life (pointing in the same direction!). In this example (activity+sleep), a linear (assumed positive) association with quality of life is assumed (higher minutes assume either healthier behavior; or non-healthier behavior, in general), but this does not address the guidelines (e.g., recommendations of 9-11 or 8-10 hours a day). What happens when someone sleeps 13 hours a day, the person still gets a higher score in the exposure variable but this non-linear relationship between sleep and health is not addressed (such high values would actually mean that someone shows a poor sleeping pattern). So, the relationship, even though continuous, does not always point to the same direction. Another example for screen time + physical activity. Assuming a sum of minutes, the coefficient of 0.10 does not tell us what we want, namely ‘more healthy minutes in those two behaviors (higher activity, less screen time) relates to more quality of life’, because the variables are reversed. A teenager with a value of 10 hours can do 8 hours of screen time and 2 hours of activity, while a teenager with 11 hours (which is now a higher value in the regression line; and we assume that this is uniformly better/worse) can do 9 hours of screen time / 2 hours of activity (worse) or 7 hours of screen time / 4 hours of activity (better). So, even though the values are increasing from 10 to 11 hours, we cannot be sure that the direction is the same. Hence, in my opinion the coefficients are not interpretable. Perhaps a different statistical approach (e.g., using combinations of ‘meeting two or more guidelines vs not’ instead of summarized minutes; compositional data analysis) should be implemented. However, I am happy to discuss the rationale of using summarized minutes in the present study. Secondly, a very brief point, I think testing whether the effect is different for boys and girls requires the inclusion of interaction terms in the main regression models. This will test whether the slopes are different rather than just showing separate results for boys and girls (one is then tempted to look at the p values and compare them, which should not be done).
Answer: Dear editor, the authors thank you for raising these questions to us as they give us the opportunity to defend the results. In this regard, it should be noted that the link to the resource you refer to is not allowed or is wrong. Thus, regarding their first question, for the formation of the combinations we have based Guimaraes et al., (2020) where through linear regression models they predicted the health indicators. However, these authors segmented the participants according to whether they met the recommendations or did not meet the recommendations for movement behavior. In this regard, the instruments used in our study would not allow us to assess the intensity of the practice time dedicated to the different types of physical activity.
Likewise, our approach has been different, we have not segmented into the population that complies with 24h Movement behavior and the population that does not comply with 24h Movement behavior, since we think that participants would be lost. In this sense, we have influenced the weight / importance of each variable in predicting quality of life, instead of assessing whether or not the participants meet the criteria. Thus, our approach allows us to assess which of the three variables is the most important for predicting quality of life. Therefore, in relation to our results, we can interpret that for a population of adolescent schoolchildren, the most important thing is the performance of physical activity as an individual behavior adjusted according to sex, BMI, age, and socioeconomic status. However, if we do not take into account the adjustment for covariates, the combination of the three behaviors predicts quality of life.
Contribution of Reviewer Discussion and Conclusion
It could be important to highlight potential measurement problems for the main exposure variables. The collected information is based on self-reports, and you could acknowledge some problems with it (e.g., reporting errors when summing up information from the past 7 days, social desirability, problems in recalling lower intensity activities, telescoping, e.g. https://pubmed.ncbi.nlm.nih.gov/33536193/).
Related to that, you could also discuss how these methodological issues may relate to problems in the estimation of effects. For example, in Table 2, the beta estimates for screen time and physical activity are very similar (0.10, 0.11) but show p values just above or below the cut off (‘dancing around the threshold’). Labeling the one as an important influence and the other as not is tricky. So, you could discuss how any shortcomings of the study (e.g., measurement, representativeness of the sample, residual confounding due to missing information when including only one movement behavior in the model) could have impacted the results (e.g., including the width of the confidence intervals, reflecting some uncertainty in the effect estimates). Mentioning/discussing these issues is important, especially concerning the ‘non-significant’ results (e.g., you highlight that the main finding was the estimate of 0.11 for physical activity, line 157).
Answer: We appreciate your recommendations. In this sense, we have changed the sentence adding some limitations:
Page 7, line 218-221 “Despite this, the study has some limitations, such as its design, which does not allow establishing cause-effect relationships, the sample size, which does not allow the results to be extrapolated to other populations, and the measurement instruments based on self-reports, which do not allow an exact assessment of the results but rather close to reality.”
Contribution of Reviewer p. 4, line 154. As mentioned above, you may consider using the wording individual and combined distributions rather than ‘independent’. Independent could also mean adjusted for the others in the regression model.
Answer: Thanks for your appreciation.The authors agree to their purpose and have modified individual and combined, rather than "independent" (page. 4, line 164)
Contribution of Reviewer p. 5, line 163. What are the differences between quality of sleep and sleep time, since your study addressed the time but here you refer to previous studies showing similar results concerning sleep quality?The conclusion could be slightly revised so that it aligns with the main results of the study. In my opinion, the conclusion should rather stick to the main findings. For example, much information about promotion and eating behavior is provided but this was not addressed by the present study.
Answer: Dear editor, we appreciate the opportunity to discuss the differences between quality and duration of sleep. In this regard, sleep quality and sleep time could be related “(Lou et al., 2012) Relation of sleep quality and sleep duration to type 2 diabetes: a population-based cross-sectional survey”. However, sleep could be considered from a holistic perspective, encompassing characteristics such as latency, time, quality, and efficiency of sleep “Development of the Sleep Quality Scale (Yi et al., 2006)”. Likewise, we do not give special importance to eating behavior, but we talk about promoting healthy habits, especially the importance of time of physical activity throughout the day for the prediction of quality of life

Reviewer 2 Report
Title: Quality of life associated health-related behaviors in High School Spanish adolescents: Promotion from schools through active methodologies
Line 32: The meaning of definition of high physical activity is missing.
Line 42: The term “higher level of physical activity” was used. What are the differences between “high physical activity” and “higher level of physical activity”?
Line 48, 50, 52, 64: The term “movement behaviours” was used without any explanation or definition.
Line 48 mentioned the literature about the relationship between movement behaviors and health-related quality of life. Line 62 stated the purpose of study was very similar with literature. What are the rationale of the study or significant of study?
Line 152: Table 2 mentioned some of the models without any explanation such as unadjusted physical activity, adjusted physical activity.
Line 219: The gender differences in both results and discussion section could be observed with significant quantity. However, no related conclusion could be made at the end.
Author Response
Dear editor,
First of all, we want to thank the reviews made by the reviewers, as it improves the quality of this article. In this sense, we believe that as a result of these modifications our article is more robust. Below you will find the changes made according to the recommendations of each reviewer.
Reviewer #2
Revisor 2
Title: Quality of life associated health-related behaviors in High School Spanish adolescents: Promotion from schools through active methodologies
Reviewer contribution: Line 32: The meaning of definition of high physical activity is missing.
Answer: dear reviewer thanks for your comment. The authors appreciate the opportunity to express their opinions to the reviewer
In this regard, we decided not to define what physical activity is, since when we go deeper into the introduction and we arrive at the concept of movement behavior, which gives us an idea of the concept of physical activity. However, if you prefer that we conceptualize it in the text, the authors agree with the decision.
Attached below the definition of physical activity to establish similarities with the concept of movement behavior
“Physical activity is defined as any bodily movement produced by skeletal muscles that results in energy expenditure.” (Caspersen et al., 1985)
Reviewer contribution: Line 42: The term “higher level of physical activity” was used. What are the differences between “high physical activity” and “higher level of physical activity”?
Answer: We appreciate your comment. In this regard, considering Poitras et al. (2016) differences as such would not exist. However, there is a nuance in terms of intensity and time in the practice of physical activity. in this sense, high physical activity would refer to the total time we do physical activity. Likewise, the level would refer to the intensity of the practice of physical activity (low, moderate or vigorous). For this reason, poitras in their findings highlights the importance of completing 60 min of physical activity in children and adolescents, but also highlights different benefits according to the types of intensity in practice.
Reviewer contribution: Line 48, 50, 52, 64: The term “movement behaviours” was used without any explanation or definition.
Answer: We appreciate your appreciation. and we have added conceptualization regarding the behavior of the movement
Reviewer contribution: Line 48 mentioned the literature about the relationship between movement behaviors and health-related quality of life. Line 62 stated the purpose of study was very similar with literature. What are the rationale of the study or significant of study?
Answer: We appreciate your opportunity to argue this question. In this regard, the authors, when we mention in the manuscript the relationship between quality of life and physical activity, we also emphasize that this topic has been little studied. Likewise, the author himself Sampasa-Kayinga et al. (2017) among his proposals for the future highlights the importance of more studies to discover these differences. For this reason, this has been the reason for the present study.
Reviewer contribution: Line 152: Table 2 mentioned some of the models without any explanation such as unadjusted physical activity, adjusted physical activity.
Answer: Thank you very much for your contribution. In this regard, when the authors refer to the term adjusted or unadjusted, we mean that the regression models are adjusted for the study covariates of sex, bmi, socioeconomic status and age.
Reviewer contribution: Line 219: The gender differences in both results and discussion section could be observed with significant quantity. However, no related conclusion could be made at the end.
Answer: thanks for your comment. In this regard in the discussion (line 177-207) the authors discuss the findings. In this sense, in the unadjusted models there were differences since for the boys only the practice of physical activity was relevant with respect to quality of life. On the contrary, in the female gender the sum of the three behaviors predicts the quality of life. In the case of the models adjusted for the covariates, only physical activity in boys predicts a better perception of their quality of life.

Reviewer 3 Report
The main purpose of the work was to analyze the association between health-related quality of life with individuals and combinations of movement behaviors, and to identify gender differences in this association.
The paper is generally well written based on sound literature, the results well presented and discussed with respect to the literature.
The work is written in an intelligible language, in accordance with the steps of the scientific method.
I have several aspects that need further attention:
The Abstract must be improved, with a sequence of the following systematization: Objectives, Methods, Results, and Conclusions.
The topic may not is very original. It would be interesting to compile the same studies during and after the pandemic.
In the introduction, the issues related to the influence of the sedentary lifestyle of children and adolescents on obesity, reduced physical fitness and mental condition can be elaborated on. It is also worth mentioning the concept of health-related fitness. The description should also pay more attention to the differences between boys and girls. Similarly, in the discussion.
If possible, provide demographic data on the research material, possibly compile in a table along with the size of the study group.
Table 1 abbreviations used are not commonly, you will can explain below the table.
Conclusions section.
In my opinion, the conclusions should be more specific. not generalized, but this is only a suggestion.
I suggest to the authors that this section be placed in paragraph form.
The application conclusion are very valuable and meaningful, although they do not fully result from the research carried out. Many biological and psychological aspects have not been studied, e.g. stress, genetics, nutrition.
The article is generally valuable and correctly written, please treat the above comments only as suggestions.
Author Response
Dear editor,
First of all, we want to thank the reviews made by the reviewers, as it improves the quality of this article. In this sense, we believe that as a result of these modifications our article is more robust. Below you will find the changes made according to the recommendations of each reviewer.
Reviewer #3
The main purpose of the work was to analyze the association between health-related quality of life with individuals and combinations of movement behaviors, and to identify gender differences in this association.
The paper is generally well written based on sound literature, the results well presented and discussed with respect to the literature.
The work is written in an intelligible language, in accordance with the steps of the scientific method.
I have several aspects that need further attention:
Contribution of Reviewer. The Abstract must be improved, with a sequence of the following systematization: Objectives, Methods, Results, and Conclusions.
Answer: Dear reviewer, we appreciate your contribution, as your recommendations improve the quality of the manuscript. In this regard, the abstract has been substantially modified
Contribution of Reviewer. The topic may not is very original. It would be interesting to compile the same studies during and after the pandemic.
Answer: We appreciate your opinion and it seems to us a very interesting question considering how the pandemic has changed our habits. However, the data collection for this study took place before the pandemic, so the purpose of our study was not to assess the impact of the pandemic on healthy habits.
Contribution of Reviewer. In the introduction, the issues related to the influence of the sedentary lifestyle of children and adolescents on obesity, reduced physical fitness and mental condition can be elaborated on. It is also worth mentioning the concept of health-related fitness. The description should also pay more attention to the differences between boys and girls. Similarly, in the discussion.
Answer: Dear reviewer regarding your question about fitness related health and specifically movement behaviors, we have not found resources that allow us to discuss this association. However, adherence to the guidelines that define correct movement behavior (, accumulate 60 minutes of moderate vigorous activity a day, 7 hours light physical activity, no more than 2 hours of screen time, and no sitting for long periods of time) has often been individually associated with fitness (Cabañas-Sánchez et al., 2018; Judice et al., 2017). Likewise, regarding gender differences in movement behavior related to the perception of quality of life, we have deepened them by adding the following information in the text [19]. "Specifically, the relationship between all three movement behaviors and health-related quality of life was significant only for interaction of the three behaviors in girls, whereas the interaction of the three behaviors in, boys points in the same direction, but compared to the girls this was not significant. In addition, a greater importance was observed in relation to quality of life for those participants who especially complied with the recommendations for sleep and sedentary time "
Júdice, P. B., Silva, A. M., Berria, J., Petroski, E. L., Ekelund, U., & Sardinha, L. B. (2017). Sedentary patterns, physical activity and health-related physical fitness in youth: a cross-sectional study. International Journal of Behavioral Nutrition and Physical Activity, 14(1), 1-10.
Contribution of Reviewer. If possible, provide demographic data on the research material, possibly compile in a table along with the size of the study group.
Answer:Thank you very much for your feedback. In this regard, we are unable to understand whether he is referring to data such as sample size, mean age of the sample, and the proportion of boys and girls in the study. In this sense, these data can be found in the text in the method section
Contribution of Reviewer. Table 1 abbreviations used are not commonly, you will can explain below the table.
Answer: We appreciate your suggestion for improvement. In this regard we have removed the abbreviations and added explanatory notes below the table
Conclusions section.
Contribution of Reviewer. In my opinion, the conclusions should be more specific. not generalized, but this is only a suggestion.
I suggest to the authors that this section be placed in paragraph form.
The application conclusion are very valuable and meaningful, although they do not fully result from the research carried out. Many biological and psychological aspects have not been studied, e.g. stress, genetics, nutrition.
The article is generally valuable and correctly written, please treat the above comments only as suggestions.
We appreciate your suggestion for improvement in this regard, a sentence has been added that specifies the conclusion reached in the study:
Answer: “The present work concludes on the importance of healthy behaviors throughout the day for a better quality of life in adolescents. Likewise, the incidence of physical activity on quality of life is highlighted as the main behavior in predicting quality of life for a population of school adolescents.”

Round 2
Reviewer 1 Report
Dear authors,
I am afraid that I have still doubts about the applied analytical approach as I think the regression coefficients of variables like PA+sleep+screen time cannot be interpreted. I also missed the interaction test when testing differences between boys and girls. Maybe you missed my previous comment about it?
Also, some minor points.
Please double-check that you report the 24h movement guidelines correctly (is there a recommendation for 7h of light PA? The sleep duration seems to be the one for the adolescents).
I also suggest providing the coefficients for both validity and reliability of all self-report measures in the text.